# Identification of Nanoparticle Properties for Optimal Drug Delivery across a Physiological Cell Barrier

**DOI:** 10.3390/pharmaceutics15010200

**Published:** 2023-01-06

**Authors:** Aisling M. Ross, Rachel M. Cahalane, Darragh R. Walsh, Andreas M. Grabrucker, Lynnette Marcar, John J. E. Mulvihill

**Affiliations:** 1BioScience and BioEngineering Research (BioSciBer), Bernal Biomaterials, Bernal Institute, University of Limerick, V94 T9PX Limerick, Ireland; 2School of Engineering, University of Limerick, V94 T9PX Limerick, Ireland; 3Department of Biomedical Engineering, Thoraxcenter, Erasmus MC, 3000 CA Rotterdam, The Netherlands; 4Health Research Institute, (HRI), University of Limerick, V94 T9PX Limerick, Ireland; 5Department of Biological Sciences, University of Limerick, V94 T9PX Limerick, Ireland; 6Department of Applied Science, Technological University of the Shannon, V94 EC5T Limerick, Ireland

**Keywords:** nanocarriers, cell barrier, characterization, permeability, toxicity, disruption

## Abstract

Nanoparticles (NPs) represent an attractive strategy to overcome difficulties associated with the delivery of therapeutics. Knowing the optimal properties of NPs to address these issues could allow for improved in vivo responses. This work investigated NPs prepared from 5 materials of 3 sizes and 3 concentrations applied to a cell barrier model. The NPs permeability across a cell barrier and their effects on cell barrier integrity and cell viability were evaluated. The properties of these NPs, as determined in water (traditional) vs. media (realistic), were compared to cell responses. It was found that for all cellular activities, NP properties determined in media was the best predictor of the cell response. Notably, ZnO NPs caused significant alterations to cell viability across all 3 cell lines tested. Importantly, we report that the zeta potential of NPs correlates significantly with NP permeability and NP-induced changes in cell viability. NPs with physiological-based zeta potential of −12 mV result in good cell barrier penetration without considerable changes in cell viability.

## 1. Introduction

In the current era of drug delivery, solubility and permeability are considered major limiting factors for bioavailability and efficacy [1]. Although attempts are being made to improve the solubility of drugs [2], permeation of these therapeutics across biological barriers to their site of action remains challenging. In order to reach the site-of-action, drugs must overcome a number biological barriers, including, not but limited to, the gastrointestinal barrier, epithelial barriers of the blood vessels, etc. [3]. The transport of drugs across these barriers proceeds via two routes: transcellular, favoured by large, hydrophobic drugs, and paracellular, the main route for small hydrophilic molecules [3]. Nanoparticle (NP) drug carriers are being investigated as a potential strategy to overcome challenges in the transport of drugs across biological barriers [4].

Many different types of NPs are currently being investigated for a variety of drug delivery applications, such as, but not limited to, inorganic NPs, polymeric NPs, and liposomes [5,6]. Inorganic NPs, such as metals, show promise in drug delivery [4,5]. These NPs can be synthesized to relatively small sizes (<100 nm), resulting in a high surface area for functionalization compared to the same mass of micron-scale particles [7]. Additionally, inherent material valances mean that these NPs are readily functionalized with ligands to enhance delivery across biological barriers and therapeutics for disease diagnosis and treatment [8,9]. Although NPs appear promising for these applications, further work is needed to increase the efficacy of drug delivery and address the non-degradative properties [5], which can lead to subsequent failure in clinical trials. However, identifying NPs that can safely and effectively deliver therapeutic payloads remains complicated because of a poor understanding of how these nanocarriers interact with the host system and how specific NP properties effect this interaction.

Moreover, most studies report the properties of NPs in water or another solvent immediately following synthesis, and few investigate how the physiological properties of the NPs might relate to the in vitro or in vivo results [10,11]. Following administration, the surface of an NP interacts with proteins and biomolecules, which form a new surface known as the “protein corona” [12,13], and this new surface is known to affect how the NP interacts with the host system [10,11,13]. Previous studies have not quantitatively compared the correlation between cellular response and the NP properties measured in water vs. physiologically relevant media. Further, the research field places significant reliance on the use of in vivo models for pre-clinical evaluation of potential NP drug carriers. There is a distinct need to Reduce, Refine, and Replace (3R Principle) the use of animal models in drug and NP testing [14,15]. Hence, the development of a simple, reliable, and rapid in vitro model could improve researchers’ ability to evaluate NP-cell interactions such as permeability, cell barrier integrity, and cell viability without the need for arduous, ethically questionable, and costly animal testing. This information can then provide feedback for the NP design and development process.

This study aims to evaluate the interaction of a series of NPs prepared from different materials (gold, silver, iron (III) oxide, titanium dioxide, and zinc oxide) in a range sizes (50, 20 and 5 nm) with a model of a cell barrier [16]. NP permeability, NP-induced changes to cell barrier integrity, and alterations to the viability of neurovascular cells will be investigated. The results of these investigations will be compared to NP properties (Zeta potential, NP size, and NP concentration) determined in water and in media [17] to establish if either of these characterisation methods is capable of predicting NP-cell interactions. This could provide leverage for improving the design of NPs based on how NPs with specific properties are predicted to perform in vivo.

## 2. Materials and Methods

An illustrative overview of the experimental procedure can be found in Figure 1.

### 2.1. Nanoparticles Synthesis and Acquisition

Gold (Au) nanoparticles (sizes 5 nm, 20 nm and 50 nm) and silver (Ag) nanoparticles (sizes 20 nm and 50 nm) were purchased from Applied Nanoparticles, Spain. Silver nanoparticles (size 5 nm) were purchased from NanoComposix, USA. All other nanoparticles (iron oxide—Fe_2_O_3_, titanium dioxide—TiO_2_, and zinc oxide—ZnO) were synthesized in-house as described in our previous work [17] and briefly detailed in Appendix A.

### 2.2. Nanoparticle Characterisation

All nanoparticles were characterised by transmission electron microscopy (TEM) and dynamic light scattering (DLS) in water and cell culture media as described in our previous work [17].

Briefly, all samples were sonicated for 1 h in a Branson 3800 sonicating bath. Following sonication, the NP suspensions were diluted to 100, 10, and 1 µg/mL. NP doses of 100, 10, and 1 µg/mL were selected based on a publication by Li et al. (2015) [18]. They calculated that a 10 µg/mL in vitro dose of gold NPs approximately equated to the dose of gold NPs being investigated in a clinical trial, corrected based on measured NP in vivo pharmacokinetics [18].

For TEM, 5 µL of NP solution was applied to a TEM grid (copper grid with carbon film) and dried overnight. An analysis was carried out on a Joel JEM-2100F; a minimum of 200 NPs were analysed per sample.

NP samples were further subjected to DLS using the ZetaSizer (Malvern Panalytical Ltd., Malvern, UK) to measure the hydrodynamic diameter (Z-average size), size distribution, and polydispersity index (PdI). Measurements were taken between 0 and 24 h post-preparation to examine time-dependent changes.

### 2.3. Cell Culture

Caco-2 cells were selected as an example biological barrier for testing. The cells were cultured in complete culture medium (CCM) prepared from Dulbecco Modified Eagle Medium (DMEM) supplemented with 10% Foetal Bovine Serum (FBS, Sigma Aldrich F7524, MQ400 quality checked, St. Louis, MI, USA), 1% non-essential amino acids (Sigma Aldrich M7145, MQ200 checked), 2 mM L-Glutamine (Sigma Aldrich G7513, MQ400 quality checked), 100 units/mL penicillin, and 100 µg/mL streptomycin (Sigma Aldrich P4333, MQ300 checked) at 37 °C in an atmosphere of 5% CO_2_. Once the cells reached confluency, they were detached using a 0.25% trypsin-EDTA solution and resuspended in DMEM at a density of 1 × 10^6^ cells/mL. All experiments were conducted with cells between passages 4–10.

### 2.4. Cell Barrier—Model Setup

Previous work demonstrated that a Caco-2 cell barrier model could reach in vivo resistance levels mimetic of restrictive biological barrier. Cell barriers were prepared as previously described [16]. Briefly,100 µL of a 1 × 10^6^ cells/mL Caco-2 cell suspension was seeded (1 × 10^5^ cells/transwell, 8.9 × 10^4^ cells/cm^2^) to the apical side of 0.4 µm pore, polycarbonate 12 well transwells (Fisher Scientific 10567522, Hampton, NH, USA). The cells were cultured for 4 days in complete CCM (exchanged every 2 days). On Day 4, the CCM was exchanged for serum-free culture medium (DMEM supplemented with 1% non-essential amino acids, 2 mM L-Glutamine, 100 units/mL penicillin, and 100 µg/mL streptomycin). The cells were maintained in serum-free medium (exchanged every 2 days) at 37 °C in an atmosphere of 5% CO_2_. All testing was conducted between days 10–18.

#### 2.4.1. Transendothelial Electrical Resistance

Throughout the experiment, the transendothelial electrical resistance (TEER) of the cell barrier was measured using the EVOM2 (World Precision Instruments, Sarasota, FL, USA). The resistance of a blank well (membrane with no cells) was subtracted from the measured TEER of the test well and multiplied by the membrane surface area (1.12 cm^2^) to calculate the actual resistance of the cell barrier. Only wells with a TEER greater than 2000 Ω.cm^2^ were used for permeability analysis.

#### 2.4.2. Paracellular Diffusion

Once TEER values above 2000 Ω.cm^2^ were achieved, the cells were tested using FITC-dextran (4 kDa) to determine the degree of paracellular transport [19]. The media was cleared from the basolateral and apical sides of the transwell. Next, 1.5 mL of fresh CCM was added to the basolateral side, and 0.5 mL of a 2 mg/mL FITC-dextran solution in CCM was added to the apical side. The transwells were incubated at 37 °C for 1 h, and 50 µL samples were taken from the basolateral compartment every 15 min. Sample fluorescence was read using a Synergy H1 microplate reader at a 490/525 nm excitation/emission. A standard curve was prepared using FITC-dextran standards of concentrations 0.005–3 mg/mL.

#### 2.4.3. Transcellular Transport

To test whether transcellular transport pathways were active, the cells were tested using ascorbic acid. The media was cleared from the basolateral and apical sides of the transwell. Next, 1.5 mL of PBS was added to the basolateral side, and 0.5 mL of a 0.1 mg/mL ascorbic acid solution in PBS was added to the apical side. The transwells were incubated at 37 °C for 1 h, and 50 µL samples were taken from the basolateral compartment every 15 min. Then, 20 µL of 0.02 M potassium permanganate solution (Merck Titripur 480160) was added to the samples, and the loss of absorbance was read using a Synergy H1 microplate reader at 525 nm, similar to the method described by Zanini et al. (2018) [20]. A standard curve was prepared using ascorbic acid standards of concentrations 0.0005–1 mg/mL.

#### 2.4.4. Efflux Pumps

Finally, efflux pump activity was probed using rhodamine-123 [21]. The media was cleared from the basolateral and apical sides of the transwell. Then, 0.5 mL of fresh CCM was added to the apical side of the transwell, and 1.5 mL of a 0.1 mg/mL rhodamine-123 solution in CCM was added to the basolateral compartment. The transwells were incubated at 37 °C for 1 h, and 50 µL samples were taken from the apical compartment every 15 min. Sample fluorescence was read using a Synergy H1 microplate reader at a 480/535 nm excitation/emission. A standard curve was prepared using rhodamine-123 standards of concentrations 0.0005–0.5 mg/mL.

### 2.5. Nanoparticle Permeability

Nanoparticle suspensions of 100, 10, and 1 µg/mL in CCM were prepared from stock suspensions. For permeability testing, each transwell and well was cleared of media. 1.5 mL of CCM was added to the basolateral side of the transwell. To the apical side of the transwell, 0.5 mL of a CCM-nanoparticle suspension was added. The models were maintained at 37 ℃ in an atmosphere of 5% CO_2_ for 12 h. Following the incubation, the basolateral compartment was collected. Samples were tested in triplicate across three separate experiments.

#### Inductively Coupled Plasma-Mass Spectroscopy

The collected nanoparticle samples were quantified by inductively coupled plasma mass spectrometry (ICP-MS) at BHP Laboratories, Limerick. The samples were quantified for zinc (ZnO nanoparticles), titanium (TiO_2_ nanoparticles), iron (Fe_2_O_3_ nanoparticles), gold (Au nanoparticles), and silver (Ag nanoparticles) as appropriate. The average permeation was then expressed as a percentage of the original loaded quantity, which was also confirmed by ICP-MS. All samples were tested in triplicate across triplicate experiments. For each experiment, media from the basolateral side of triplicate transwells was pooled (4.5 mL total volume). The samples were then prepared for ICP-MS analysis via acid matrix digestion. Then, 1 mL of sample was added to 49 mL of acid matrix. Acid matrix consisted of 6.7% HNO_3_ and 1.2% HCl in deionised water. Further details of the ICP-MS analysis can be found in Appendix A below.

### 2.6. Cell Barrier Integrity

#### 2.6.1. Transendothelial Electrical Resistance

Following nanoparticle incubation, three TEER readings were taken per transwell and compared to the readings taken before nanoparticle administration (Section 2.4.1). A 2 h incubation with H_2_O_2_ acted as a positive control.

#### 2.6.2. Immunofluorescence

A 28.6 µL of a 1 × 10^6^ cells/mL Caco-2 cell suspension was seeded in 96 well plates (2.86 × 10^4^ cells/well, 8.9 × 10^4^ cells/cm^2^). The cells were cultured as in Section 2.3. On Day 10, each well was cleared of media and 160 µL of nanoparticle suspension (100, 10, or 1 µg/mL) was added to each well. The models were maintained at 37 °C in an atmosphere of 5% CO_2_ for 12 h. A 2 h incubation with H_2_O_2_ acted as a positive control. Following incubation, the nanoparticle suspension was removed, and the cells were fixed with 4% paraformaldehyde for 15 min, permeabilised with 0.5% Triton X-100 for 15 min, and blocked with 10% FBS, 0.0005% Triton X-100 for 1 h. The wells were incubated with either rabbit anti-occludin (Sigma Aldrich SAB3500301, 1:200, MQ100 quality checked) or rabbit anti-ZO-1 (Biosciences 61-7300, 1:200, Durham, NC, USA) overnight at 4 °C followed by 1 h incubation with CF 555-conjugated goat anti-rabbit (Sigma Aldrich SAB4600069, 1:1000, MQ100 quality checked) at room temperature. The cells were incubated at room temperature with DAPI (D9542-10MG, 0.1 µg/mL) for 10 min. The wells were imaged using the Imagexpress Micro Confocal High-Content Imaging System (Molecular Devices). Then, 9 different images were acquired per well, and triplicate wells were tested for each experimental conditions across three separate experiments.

### 2.7. Cell Toxicity—MTT Assay

Cells were cultured each as described in Section 2.3. 28.6 µL of a 1 × 10^6^ cells/mL Caco-2 cell suspension was seeded in 96 well plates (8.9 × 10^4^ cells/cm^2^). On Day 10, each well was cleared of media, and the NP solution was added. The cells were incubated with nanoparticle solution for 1, 4, 12, and 24 h. 15 min incubation with 0.5% Triton X-100 in PBS was used as a positive control,

After incubation, the nanoparticle solution was removed from the cell cultures. Then, 100 µL 0.5 mg/mL 3-[4,5-dimethylthiazol-2-yl]-2,5-diphenyltetrazolium bromide (MTT) solution (Fisher Scientific Ireland 15234654) in serum-free media was added to each well and incubated for 2 h at 37 °C. Then, 100 µL of solubilisation solution (5 g of SDS in 50 mL of 0.01 M HCl) was added to the wells and incubated for a further 4 h at 37 °C. The absorbance was then read on a Synergy H1 microplate reader at 570 nm. A well with no cells was used as an MTT control. Cell viability is expressed in Equation (1).
(1)$$\frac{\mathrm{Sample}\;\mathrm{Absorbance}\;-\;\mathrm{MTT}\;\mathrm{Control}\;\mathrm{Absorbance}\;}{\mathrm{Untreated}\;\mathrm{Control}\;\mathrm{Absorbance}}\times 100$$

### 2.8. Statistical Analysis

All statistical analysis was carried out using Prism 6 (version 8.3.1, serial number GPS-1552720-LHRD-469E4, Graphpad, San Diego, CA, USA). Any outliers were identified using Grubb’s outliers test (Q = 1) and removed from the dataset. The mean permeability of each nanoparticle was compared using a Kruskal-Wallis test with Dune’s multiple comparison test. The mean change in TEER for each nanoparticle was compared to the negative control (in CCM) using a Kruskal-Wallis test with Dune’s multiple comparison test. For toxicity analysis, samples were compared to the negative control (in CCM) using a 2-way ANOVA (ANOVA) with Dunnett’s multiple comparison test.

Correlations between NP permeability, TEER, or toxicity data, and the NP properties in water or media were investigated using the Pearson’s and Spearmen’s correlation. For all tests a 95% confidence interval was used and statistical significance is reported as a *p* < 0.05 (* = 0.01 < *p* < 0.05; ** = 0.001 < *p* < 0.01; *** = 0.0001 < *p* < 0.001; **** = *p* < 0.0001).

## 3. Results

### 3.1. Cell Barrier Permeability Assay

#### 3.1.1. Paracellular and Transcellular Transport

Conjugated to a FITC molecule, 4 kDa Dextran was utilised to investigate paracellular transport. Here, we found that the permeability coefficient for 4 kDa Dextran across the cell barrier (at 15 min) was 4.09 ± 1.32 × 10^−7^ cm/s, indicating that a restrictive barrier was formed with low levels of paracellular diffusion [10].

Next, transcellular transport was investigated. Ascorbic acid is taken up by endothelial cells in its oxidised form (dehydroascorbic acid) by glucose transporters [22]. Once inside the cell, it is reduced to ascorbic acid again [23]. The quantity of ascorbic acid crossing the cell barrier model was estimated by measuring the ascorbic acid-based reduction of potassium permanganate. The permeability coefficient was found to be 6.72 ± 4.84 × 10^−4^ cm/s (at 15 min), similar to reported values of 0.05 cm/h [24]. This indicates that transcellular transport is active in the cell barrier model.

Finally, the activity of efflux pumps was verified; a significant number of therapeutic agents do not reach the site of action due to the activity of efflux pumps in areas such as the gastrointestinal tract, the blood brain barrier, and the blood-tumour barrier [25,26]. Rhodamine-123, a substrate of P-glycoprotein, was used to evaluate efflux pump activity in the cell barrier model [25]. The permeability coefficient for efflux of rhodamine-123 was calculated to be 1.17 ± 0.50 × 10^−5^ cm/s (at 15 min) similar to reported values between 6 × 10^−4^ and 2.5 × 10^−6^ cm/s [27,28,29].

#### 3.1.2. Nanoparticle Permeability

The transport of spherical Au, Ag, Fe_2_O_3_, TiO_2_, and ZnO NPs of sizes 50, 20, and 5 nm across the cell barrier model were tested at concentrations of 100, 10, or 1 µg/mL. The permeation of the NPs across the cell barrier model was expressed as a percentage of the original dose and is reported in Figure 2a–e. In general, the 1 µg/mL dose showed the greatest permeation (as a percentage of the original dose).

Based on the materials investigated, Fe_2_O_3_ and ZnO NPs showed the greatest levels of permeation across all sizes (Figure 2c,e). The levels of permeation of Ag were lower (b) while the quantity of TiO_2_ and many of the Au NPs were undetectable (Figure 2a,d). Any permeation of these NPs fell below the limit of quantification for the ICP-MS analysis (Appendix A), which is one of the most sensitive techniques that can be utilised to quantify NP permeability [11]. Graphs of the quantity of NP permeate can be seen in Appendix A.

We then investigated the relationship between NP permeability and the measured NP characteristics (i.e., size and ζ-potential [17]). It was found that NP size did not predict the ability of the NPs to cross the cell barrier model (Appendix A). Instead, we found that NP ζ-potential, characterised in water, was the best predictor of cell barrier permeation with a *p*-value less than 0.0001 and a Spearman’s correlation coefficient of 0.5828 (Figure 2f). The more negative the charge on the surface of the NP, the lower the level of permeation. As the NP surface charge approaches 0 or positive values, permeation is improved.

NP ζ-potential in media was also significantly correlated with NP permeation, although less significantly than the water-based ζ-potential (*p* = 0.0027, ρ = −0.4374) (Figure 2g). Water-based and media-based ζ-potential are likely related to each other. A positive, water-based ζ-potential results in NP association with negatively charged proteins. This results in a more negative media-based ζ-potential. Meanwhile, the opposite is seen for NPs that have negative water-based ζ-potentials.

### 3.2. Cell Barrier Integrity

The permeation of NPs across the cell barrier could be the result of disruption to cell barrier tight junctions. Although transient, controlled loosening of tight junctions in epithelial and endothelial barriers could improve paracellular diffusion and drug delivery, uncontrolled penetration of substances across cell barriers via this route could have negative physiological effects [30,31]. Figure 3 shows the fold change in TEER values relative to the TEER value before NP incubation. It was found that none of the NP samples or concentrations tested resulted in a significant change in barrier integrity. In most incidences, the 100 µg/mL doses of 5 nm particles resulted in an insignificant reduction in TEER. Since no significant change in TEER was observed, there was no significant correlation between NP properties and change in TEER (Appendix A).

### 3.3. Cell Toxicity Assay

Viability assays on the Caco-2 cells were then carried out. The Caco-2 cells were tested at 4 time points within a 24 h period (1, 4, 12, 24 h) to determine if a time-dependent toxic response was noted. As seen in Figure 4, high concentrations of ZnO NP resulted in the greatest change in cell viability, displaying a highly statistically significant increase in mitochondrial activity at the early time points, which decreased over the course of the experiment. This is potentially the result of the water-soluble ions in ZnO [32,33,34]. It has been established that zinc ions from ZnO can lead to reactive oxygen (ROS) generation and mitochondrial damage [34]. This increase in metabolic activity is observed in the lower concentrations at the 24 h time point. A significant increase in mitochondrial activity was also noted for the 5 nm Ag samples, the 50 and 20 nm TiO_2_ and the 5 nm Fe_2_O_3_ samples, all at concentrations of 100 µg/mL. Interestingly, no significant reductions in mitochondrial activity were noted across all samples, concentrations, and time points.

When the cellular response was correlated to NP properties, it was found that NP ζ-potential in media was the best predictor of NP-induced changes in cell viability. For the Caco-2 cells, media-based ζ-potential significantly correlated to changes in cell viability for the 4 and 12 h samples with *p* values of 0.0401 and 0.043, respectively (Figure 4g,h). ζ-potentials between −10 and −12 mV in media resulted in MTT viability assay results close to 1.

## 4. Discussion

The field of nanomedicine has been rapidly expanding in recent decades. Despite interest in NPs, the translation of NPs to the clinic has remained low. During clinical trials, these materials do not perform as expected. This results in a considerable cost to research groups and pharmaceutical companies in addition to a significant loss of time. An improved understanding of NP material properties that result in favourable physiological outcomes could represent a means of addressing this bottleneck to translation from pre-clinical to bed-side discovery [35]. If the physiological response could be estimated based on NP properties, NP design could be tailored to create NPs that are anticipated to perform optimally in vivo.

Although previous research has been conducted to address the potential for predicting the physiological responses based on NP properties, few have conducted comparisons between multiple materials and sizes. Further, to the best of our knowledge, the relationship between NP properties, determined in physiological fluid and their interaction with the cell barrier, has not been investigated previously to the same extent as reported here. Following administration, NPs typically experience considerable adsorption of proteins onto the NP surface. This new surface will interact with cells in the body and alter the fate of the NPs [6,17]. Therefore, we sought to investigate if NP properties, determined in physiological fluid, were a better predictor of NP-cell barrier interactions than those properties determined in water.

We have previously shown a significant difference in NP properties when water-based and media-based dispersions are compared [17]. In media, there are significant changes to NP size and ζ-potential. ZnO and Ag NPs samples show a reduction in hydrodynamic diameter, suggesting possible NP degradation. Conversely, Fe_2_O_3_ and TiO_2_ particles tend to increase in size, potentially due to agglomeration or significant protein corona formation. In water, NP ζ-potentials span a broad range of values (−65 to +20 mV). Meanwhile, in media, ζ-potential ranges from just −5 to −15 mV. Unfortunately, the results of serum-based dispersion were inconclusive. This was due to significant interference from the high concentration of proteins in the suspension [17]. Therefore, media-based dispersion may be more useful for estimating in vivo physiological properties than pure serum suspensions.

One of the most common characteristics reported in NP studies is, unsurprisingly, the size of the NP. It is generally considered that the smaller the NP, the better the delivery across cell barriers in vivo [36]. However, size is also believed to be related to toxicity, with smaller particles being more toxic than larger ones [37]. Therefore, a delicate balance exists between improving permeability whilst simultaneously reducing toxicity.

Here, we investigated the relationship between NP size and cell barrier interactions, including permeation, barrier disruption, and toxicity. No significant correlation was detected for water- or physiological-based NP size and the cell barrier interactions (Appendix A), although nearly significant correlations were seen between NP size and the 1 and 24 h Caco-2 viability (Figure 4f,i). It has been reported that smaller NPs induce more toxic responses than larger NPs [38,39]. It is thought that the smaller sizes can lead to greater cell internalisation leading to membrane damage and reactive oxygen species generation [39]. However, it is also reported that cell viability is independent of size for other materials [40].

Regarding cell barrier permeability, we did not observe a significant effect of size on the permeability of NPs across the barrier model. The benefit of decreasing particle size for enhanced cell barrier penetration may be more applicable when comparing NPs of greater sizes [36,41]. It has been reported that NPs in the range of 25 to 100 nm show better permeability than those of sizes greater than 100 nm [36,42]. However, the scale of NPs tested here ranges from 3 to 60 nm. Potentially, at this scale, the penetration of NPs becomes more dependent on other NP properties, such as surface chemistry.

In contrast to the lack of correlation between NP size and cell barrier interactions, ζ-potential was found to significantly correlate to cell barrier permeability and cell viability.

For the cell barrier permeability data, both ζ-potential measured in water and media correlated significantly to the ability of NPs to cross the cell barrier. It has previously been suggested that negative ζ-potential is required for passage through a cell barrier [43]. However, Voigt et al. (2014) reported that both positive and negatively charged NP are capable of permeating across the blood-retinal barrier [35]. The results presented here are in agreement with those reported by Voigt et al. (2014). We also report that positive and negatively charged NPs cross this cell barrier model. In the study conducted by Voigt et al., only NPs of 5 mV and −26 mV were compared. Both these charges fall within the range of NPs we report that are capable of crossing the cell barrier in this study. Instead, NPs that are more negatively charged were not capable of crossing the cell barrier.

The permeability of the positively charged NPs could also be related to orosomucoid activity. Orosomucoid is a plasma glycoprotein synthesised by the liver and released into circulation [44]. Orosomucoid in serum has been shown to bind to the surface of both intestinal epithelial cells and endothelial cells [44]. Orosomucoid, which is negatively charged at the physiological pH, is believed to play a role in controlling cell barrier permeability by conferring a negative charge to the cells in the barrier. In this way, it is believed to modulate the permeability of charged materials by increasing the permeability of positively charged materials while reducing the permeability of negatively charged materials. This could account for the low permeability of highly negatively charged NPs across the cell barrier reported here.

Given that the ζ-potential of NPs in media also correlated with NP permeability, the relationship between NP ζ-potential and cell barrier permeability could be related to the formation of the protein corona. The charge of the proteins in the protein corona, which is likely dictated by the original NP ζ-potential, could also influence permeability. Highly negatively charged NPs are more likely to associate with positively charged proteins. Meanwhile, positively charged NPs will associate with negatively charged proteins, and NPs close to a neutral charge are more likely to associate with both positively and negatively charged proteins. It has previously been reported that apolipoprotein E (apo E) has been found in the protein corona of ZnO and SiO_2_ NPs [45,46]. A coating of Apo E on NP surfaces can enhance permeation of NPs across biological barriers via endocytosis and transcytosis, potentially mediated by lipoprotein receptors [47,48]. Apo E has a pI of ~5.4 and therefore exists as a negatively charged protein at the physiological pH [49]. Hence, this protein may bind to the positively charged NPs, thereby mediating their transport across biological barriers.

We also report here that the ζ-potential of NPs, measured in media, correlated with cell viability. ζ -potential has previously been implicated in nanotoxicological investigations. The toxicological response of cells to NPs can be related to the membranolytic potential of the NPs [50]. It was reported by Cho et al. (2013) that the NPs in serum tended towards a ζ-potential of approximately −10 to −12 mV, and these particles did not elicit any significant toxicological effects on the cells tested [51]. However, in protein-free conditions, the ζ-potential of the NPs correlated well with the lytic potential of the NPs [50]. Here, we see that, in general, NPs with physiological ζ-potentials that deviate from approximately −10 to −12 mV are associated with changes in cell viability. Further, Cho et al. report that when the protein corona of metallic NPs is enzymatically digested, the lytic potential of the NPs is restored [50]. Taken together, this data suggests that the formation of the protein corona, which alters NP ζ-potential, affects the membranolytic potential of the NPs. The formation of a protective protein corona can shield cells from the lytic activity of highly charged NPs [50,51]. Hence, determining the physiological-based ζ-potential of novel nanocarriers could be a good predictor of cell viability.

### Limitations

For the permeability analysis, only one time point was investigated. Therefore, it is unclear what the rate of the NP permeation is. Testing for 12 h could overestimate NP permeation if the NP would normally be cleared from the body before this time. In contrast, 12-h testing could underestimate permeation if there is prolonged retention in the body. However, this testing does facilitate direct comparisons between the permeability of different NPs.

Similarly, only one time point was investigated for the cell barrier integrity analysis. The 12-h time point was selected to correspond to the permeability analysis. This way, any elevated permeation resulting from cell barrier disruption could be identified. No significant changes in cell barrier integrity were observed during this testing. However, some samples do show reductions in TEER and these changes may become more significant if the barriers were subjected to extended NP exposure. We also did not investigate if these effects were transient or whether the cell barrier returned to pre-treatment levels once the NP solution was removed.

For cell viability testing, MTT assays were carried out. This assay only measures mitochondrial activity [52] as a measure of cell viability. Changes in cell viability can result from many distinct factors, which are not investigated in detail here. Further, damages to other cell processes may occur that do not translate to a direct change in mitochondrial activity. However, MTT assays are rapid, high-throughput colorimetric assays that enable a large number of samples, concentrations, and time points to be tested [53].

## 5. Conclusions

Advancing the field of nanomedicine could benefit greatly from an ability to predict the interaction of NPs with cells at the NP design stages. Here, we investigated the interaction of NPs at 3 different concentrations, prepared from 5 different materials with varying sizes and surface properties with a model of a biological barrier. We found that, in general, Fe_2_O_3_ NPs performed the best with good penetration across the cell barrier model and few changes to cell viability as a result. These NPs may be suitable as carriers for the delivery of therapeutics.

Importantly, we have shown that NP ζ-potential, measured in media, correlates well to both cell barrier penetration and cell viability. Future testing incorporating NPs of different shapes or materials could be compared to those presented here to investigate further correlations. Generally, NPs with a media-based ζ-potential of between −10 and −12 mV showed both good cell barrier permeability and little changes to cell viability across all cell lines. This could be related to the formation of the protein corona, with surface proteins aiding delivery whilst also shielding cells from any potential toxic effects. Media-based ζ-potential could be a valuable tool to screen NPs intended for drug delivery applications, prior to beginning any pre-clinical testing.

## Figures and Tables

**Figure 1 pharmaceutics-15-00200-f001:**
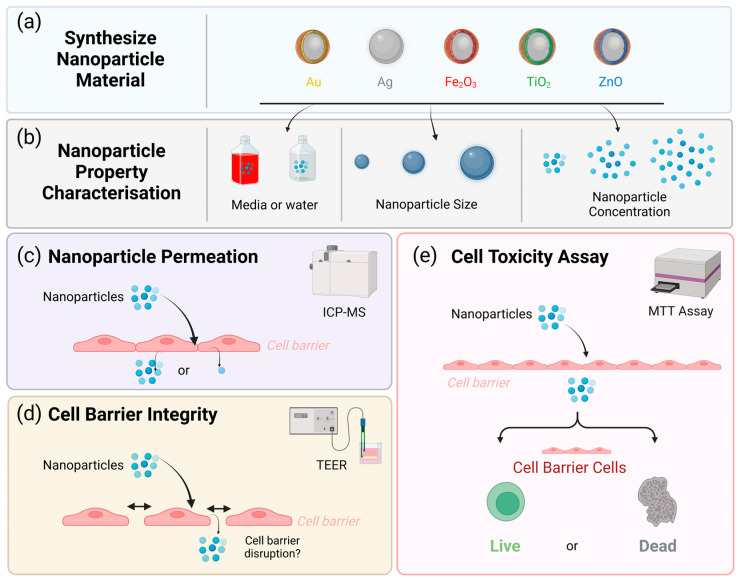
An overview of the experimental steps of this study; (**a**) 5 different nanoparticles were synthesized and (**b**) dispersed in either water or cell culture media to evaluate their properties, including size and zeta potential. These nanoparticles were then applied to an in vitro biological barrier model to assess (**c**) permeation or (**d**) nanoparticle permeation across a cell barrier using inductively coupled plasma mass spectrometry (ICP-MS), (**d**) cell barrier integrity once the nanoparticle has crossed using Transendothelial Electrical Resistance (TEER) and finally, (**e**) cell toxicity of the cell barrier (Caco-2). Created with BioRender.com—HV24LU5D3Q.

**Figure 2 pharmaceutics-15-00200-f002:**
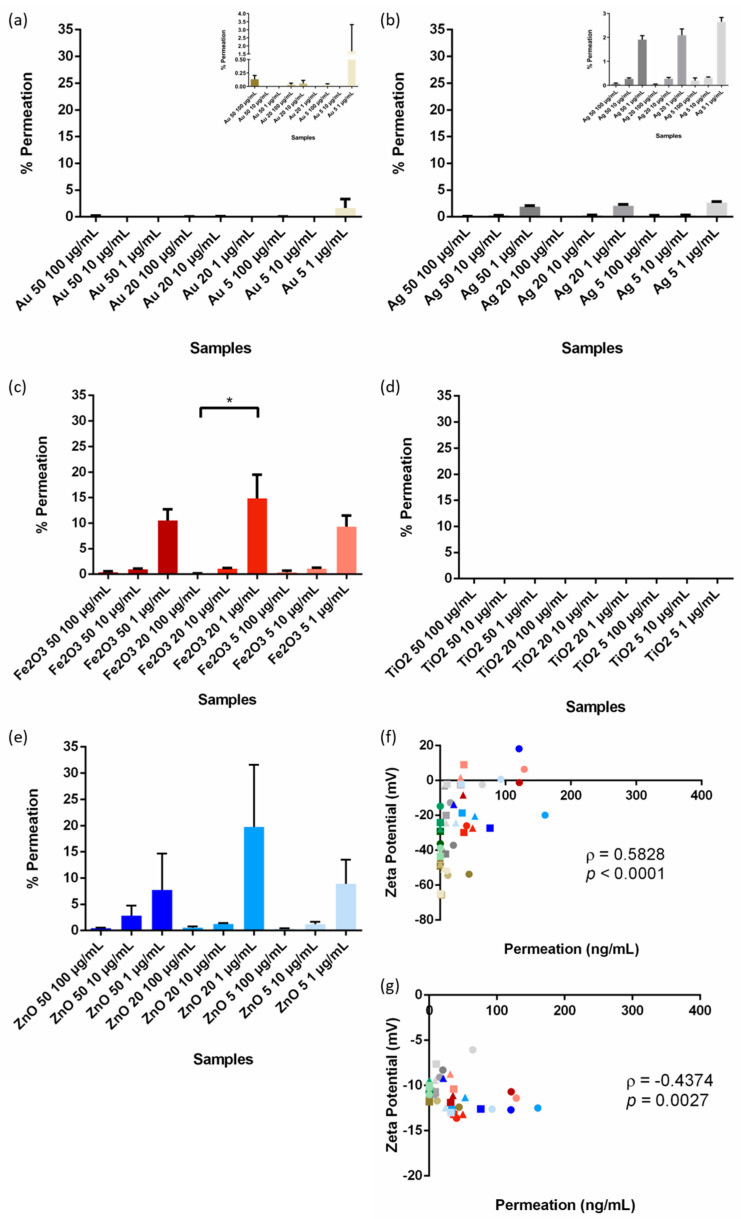
NP permeation across the cell barrier, as a percentage of the applied dose, following 12-h incubation with (**a**) gold, (**b**) silver, (**c**) iron (III) oxide, (**d**) titanium dioxide (there was no permeation of NPs found in all tests and all values found were zero i.e., the chart is blank), and (**e**) zinc oxide nanoparticles. For several samples, particularly all TiO_2_ based NPs, no material could be detected by the ICP-MS in the permeate and hence the percentage permeation is noted as 0%. (**f**) Spearman’s correlation between NP permeability and nanoparticle zeta potential in water. (**g**) Spearman’s correlation between NP permeability and nanoparticle zeta potential in media. In the scatterplots ● = 100 µg/mL dose, ■ = 10 µg/mL, Δ = 1 µg/mL, * = 0.01 < *p* < 0.05. Note: levels were almost undetectable in (**a**) and (**b**). Colour Key for all graphs: Au = Gold, Ag = Grey, Fe_2_O_3_ = Red, TiO_2_ = Green, and ZnO = Blue.

**Figure 3 pharmaceutics-15-00200-f003:**
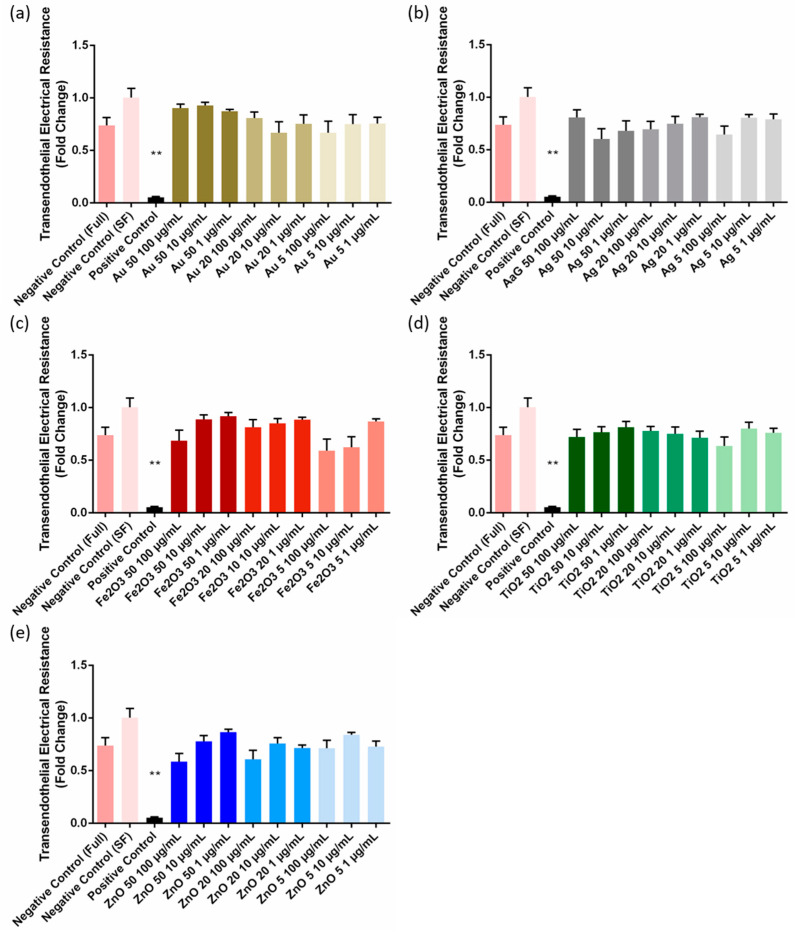
Fold change in transendothelial electrical resistance values following 12-h incubation with (**a**) gold, (**b**) silver, (**c**) iron (III) oxide, (**d**) titanium dioxide, and (**e**) zinc oxide nanoparticles of different concentrations and sizes. Colour Key for all graphs: Au = Gold, Ag = Grey, Fe_2_O_3_ = Red, TiO_2_ = Green, and ZnO = Blue. ** = 0.001 < *p* < 0.01, SF = serum free media, Full = media containing serum.

**Figure 4 pharmaceutics-15-00200-f004:**
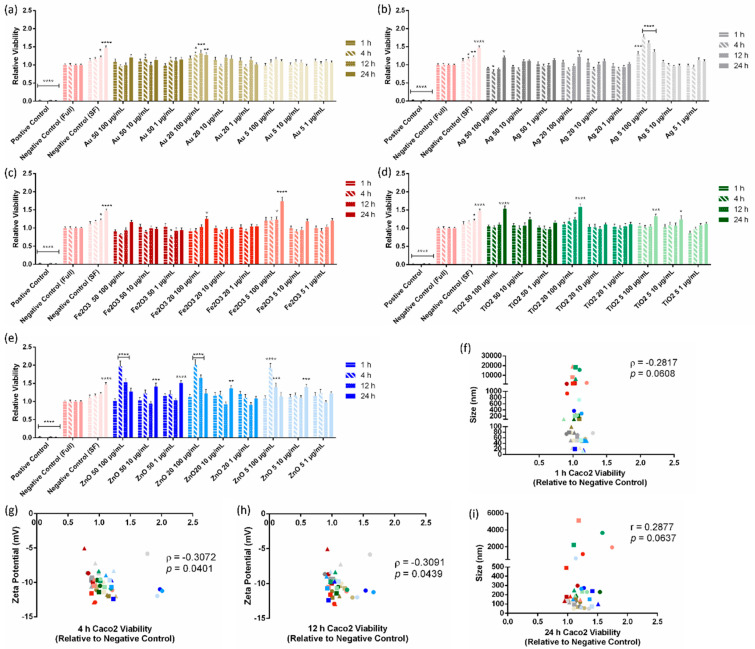
Results of MTT viability assay for Caco-2 cells treated with (**a**) gold, (**b**) silver, (**c**) iron (III) oxide, (**d**) titanium dioxide, and (**e**) zinc oxide nanoparticles. All samples were analysed at 1, 4, 12, and 24 h. (**f**) Spearman’s correlation between Caco-2 cell viability after 1 h treatment and nanoparticle size in media. (**g**) Spearman’s correlation between Caco-2 cell viability after 4 h treatment and nanoparticle zeta potential in media. (**h**) Spearman’s correlation between Caco-2 cell viability after 12 h treatment and nanoparticle zeta potential in media. (**i**) Pearson’s correlation between Caco-2 cell viability after 24 h treatment and nanoparticle size in media. Colour Key for all graphs: Au = Gold, Ag = Grey, Fe_2_O_3_ = Red, TiO_2_ = Green, and ZnO = Blue. ● = 100 µg/mL nanoparticle suspension, ■ = 10 µg/mL nanoparticle suspension, Δ = 1 µg/mL nanoparticle suspension, * = 0.01 < *p* < 0.05; ** = 0.001 < *p* < 0.01; *** = 0.0001 < *p* < 0.001; **** = *p* < 0.0001, SF = serum free media, Full = media containing serum.

## Data Availability

Not applicable, more data is presented in Appendix A.

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
