# Peer review of "Identification of Nanoparticle Properties for Optimal Drug Delivery across a Physiological Cell Barrier"

_pharmaceutics, 2023, doi:10.3390/pharmaceutics15010200_

Round 1
Reviewer 1 Report (New Reviewer)
The authors report results on the efficiency and optimization of drug delivery nanoparticle (NPs) for permeability across the cell barrier. The NPs effects on the cell barrier integrity and cell viability are evaluated respectively. The components for the NPs preparation include Au, Ag, ZnO, TiO2, and Fe2O3.
I have a few minor suggestions for correction and improvements:
1. The purity of the chemical components should be given explicitly. Usually it is written by the manufacturer on the vial label.
2. Fig. 2d is empty. Please show the necessary plot.
3. Fig. 4 panels f), g), h), and i) are squeezed in a such small space that it is impossible to see the axes. A better enlarged view will help for the readability.
Author Response
Thank you for you comments, we have addressed below and tracked in the resubmission.
- We have included, where possible, the purity of the chemicals used (mainly in Appendix A) as well as the quality control levels for the different cell culture reagents.
- Also to add we have more information regarding the methods in the Supplementary File. If the methods seem short there is more information provided in Appendix A and B that will assist the reader.
- Figure 2d is empty on purpose as these are the direct results we received from the ICP-MS instrument. We felt it correct to leave it as is otherwise it would be inconsistent not to show the results. Even if we place an inserted image, similar to panel a and b, it would still be 0. Therefore, we shall leave it in but mention in the caption to reflect that this is not an error.
- We have included in the caption "(d) titanium dioxide (there was no permeation of NPs found in all tests and all values found were zero i.e., the chart is blank), "
- Agreed, we have edited this figure to increase clarity and readability. See attached file.

Reviewer 2 Report (Previous Reviewer 4)
The authors have addressed the comments.
Author Response
Thank you for your previous comments and positive feedback.
Reviewer 3 Report (Previous Reviewer 2)
The authors had responded to all my raised comments. However, I still recommend to remove paragraphs heading in the Discussion section
Author Response
Thank you for your comments. We have removed the discussion headings as mentioned. However, we have kept one sub-heading for Limitations so readers are fully aware that we have considered the limitations of our work. See the updated manuscript file.
This manuscript is a resubmission of an earlier submission. The following is a list of the peer review reports and author responses from that submission.
Round 1
Reviewer 1 Report
Interesting study and planned scientifically well, however No evidences of selective translocation of Nanoparticles has been provided.
hence It is expected to modify the Introduction section accordingly.
Author Response
We thank the reviewer for their assessment. In relation to your comment, we agree that translocation is not studied here and that animal models would be necessary to study this. Therefore, to highlight this we have included the following text to the introduction (LINES 70-74) to clarify this:
“Although animal models will still be necessary to evaluate the selective translocation of NPs to the CNS, a realistic in vitro model will improve our understanding of the properties that allow NPs to cross biological barriers and reach the CNS. This information can then to feedback into the NP design and development process.”

Reviewer 2 Report
In this study, the authors investigated the interaction of NPs at 3 different concentrations, prepared from 5 different materials with varying sizes and surface properties with a cell barrier mimetic of the resistance found in the BBB in vivo. The study discusses a very important issue that may help in the translation of many nanoparticle formulation into clinic. However, the authors are requested to respond to the following minor comments prior the acceptance of the manuscript
1- Please correct Figure citation in the text in the Results Section.
2- The quality of Figures should be improved throughout the manuscript.
3- In Discussion Section: It is preferred to remove subheading and to make it more concise
4- Many typographical errors were found. Please check.
Author Response
1- Please correct Figure citation in the text in the Results Section.
Thank you for identifying this issue. Unfortunately, the Figure citations did not remain when submitted. I have rectified this now with all references to Figures.
2- The quality of Figures should be improved throughout the manuscript.
All figures have now been set to the required resolution of 300 DPI as per the journal’s submission rules.
3- In Discussion Section: It is preferred to remove subheading and to make it more concise
We have removed the second layer of subheadings to make the discussion section more concise (i.e. 4.3.1 and 4.3.2 so that it is clearer.
4- Many typographical errors were found. Please check.
All authors have reviewed and made the typo corrections throughout.

Reviewer 3 Report
The manuscript focuses on the identification of factors controlling the delivery across the physiological cell barrier. For this purpose the NPs prepared from 5 materials, with varying sizes and surface properties at 3 concentrations were investigated.The research is well descigned and the results clearly presented and therefore, I recommend to publish the manuscript after small improvement:
- please, check line 94-95, page 2
- from the page 7 the "Error! Reference source not found"appeares instead of number of figures - it should be corrected
Author Response
- please, check line 94-95, page 2
We have rectified this sentence structure error, please see new sentence now.
“Further detail for this characterisation is provided in Appendix B. A brief summary of the zeta potential measurement is provided below, with further detail on microscopy and DLS provided in Appendix B.”
- from the page 7 the "Error! Reference source not found"appeares instead of number of figures - it should be corrected
Thank you for identifying this issue. Unfortunately, the Figure citations did not remain when submitted. I have rectified this now with all references to Figures.

Reviewer 4 Report
The manuscript entitled “Identification of Nanoparticle Properties for Optimal Drug Delivery Across a Physiological Cell Barrier” propose a coating that can effectively inhibit the bacteria on the surface of materials. Although the manuscript is meaningful, there are some problems in it. Therefore, I recommend this paper should be published after revision.
1. The article does not show the relevant experimental mode diagram.
2. Why are the ordinates of a-plot and b-plot in Figure 1 not adjusted to an appropriate scale?
3. The d-plot in Figure 1 is for TiO2 and its value is blank, then the note under Figure 1 should explain. Otherwise, it is considered to be a blank image
4. The whole work is only at the cellular level, why no animal experimental data.
Author Response
- The article does not show the relevant experimental mode diagram.
We have now included a figure that now gives and overview of the experimental process. This has now been inserted in LINE 95 with the below figure caption as well.
Please see the attachment and the manuscript for the updated Figure 1.
Figure 1. An overview of the experimental steps of this study (a) nanoparticle materials synthesized, (b) the 3 nanoparticle properties that were altered to determine their effect on the following; (c) nanoparticle permeation across a cell barrier using inductively coupled plasma mass spectrometry (ICP-MS), (d) cell barrier integrity once the nanoparticle has crossed using TransEndothelial Electrical Resistance (TEER) and finally, (e) cell toxicity of astrocytes (DI-TNC1), neurons (SH-SY5Y) and the cell barrier (Caco-2).
- Why are the ordinates of a-plot and b-plot in Figure 1 not adjusted to an appropriate scale?
The axes were chosen to allow for direct comparison of each NP material in Figure. However, we agree that it is difficult to see the quantities in plots a and b of Figure 1. Therefore, we have added an insert to these plots showing a more appropriate axis scale for these materials.
- The d-plot in Figure 1 is for TiO2 and its value is blank, then the note under Figure 1 should explain. Otherwise, it is considered to be a blank image
No TiO2 was detected in the samples and therefore, this graph is empty but has been included for comparison to the other materials. This has been clarified in the caption as follows:
“Figure 1. NP permeation across the cell barrier, as a percentage of the applied dose, following 12-hour incubation with (a) gold, (b) silver (c) iron (III) oxide, (d) titanium dioxide, (e) zinc oxide nanoparticles. For several samples, particularly all TiO2 based NPs, no material could be detected by the ICP-MS in the permeate and hence the percentage permeation is noted as 0%. (f) Spearman’s correlation between NP permeability and nanoparticle zeta potential in water. (g) Spearman’s correlation between NP permeability and nanoparticle zeta potential in media. In the scatterplots ● = 100 µg/mL dose, ■ = 10 µg/mL, Δ = 1 µg/mL.Note: levels were almost undetectable in (a) Colour Key: Au = Gold, Ag = Grey, Fe2O3 = Red, TiO2 = Green, and ZnO = Blue.) and (b). Colour Key for all graphs: Au = Gold, Ag = Grey, Fe2O3 = Red, TiO2 = Green, and ZnO = Blue.”
- The whole work is only at the cellular level, why no animal experimental data.
The review highlights an important aspect to the limitations of this work which is to include animal work. The short answer is that we do not have any access to animal models in our institute nor was it the current focus of this work. Our project aims to develop a method to act as a triage before the animal work so that when there is an investment into animal models that a group can remove unnecessary or non-functional nanoparticles.
Therefore, to highlight this we have included the following text to the introduction (LINES 70-74) to clarify this:
“Although animal models will still be necessary to evaluate the selective translocation of NPs to the CNS, a realistic in vitro model will improve our understanding of the properties that allow NPs to cross biological barriers and reach the CNS. This information can then to feedback into the NP design and development process.”
